# Efficacy and Safety of a Ketogenic Diet in Children and Adolescents with Refractory Epilepsy—A Review

**DOI:** 10.3390/nu12061809

**Published:** 2020-06-17

**Authors:** Jana Wells, Arun Swaminathan, Jenna Paseka, Corrine Hanson

**Affiliations:** 1College of Allied Health Professions, University of Nebraska Medical Center, 984045 Nebraska Medical Center, Omaha, NE 68198-4045, USA; ckhanson@unmc.edu; 2Department of Neurological Sciences, University of Nebraska Medical Center, 988440 Nebraska Medical Center, Omaha, NE 68198-8440, USA; arun.swaminathan@unmc.edu; 3Department of Pharmaceutical and Nutrition Care, Nebraska Medicine 4350 Dewey Ave, Omaha, NE 68105, USA; jpaseka@nebraskamed.com

**Keywords:** epilepsy, seizure, nutrition, ketogenic, diet

## Abstract

Epilepsy in the pediatric and adolescent populations is a devastating condition where individuals are prone to recurrent epileptic seizures or changes in behavior or movement that is the direct result of a primary change in the electrical activity in the brain. Although many children with epilepsy will have seizures controlled with antiseizure medications (ASMs), a large percentage of patients are refractory to drug therapy and may consider initiating a ketogenic diet. The term Ketogenic Diet or Ketogenic Diet Therapy (KDT) refers to any diet therapy in which dietary composition results in a ketogenic state of human metabolism. Currently, there are 4 major Ketogenic diet therapies—the classic ketogenic diet (cKD), the modified Atkins diet (MAD), the medium chain triglyceride ketogenic diet (MCTKD) and the low glycemic index treatment (LGIT). The compositions of the 4 main KDTs differ and limited evidence to distinguish the efficacy among different diets currently exists. Although it is apparent that more randomized controlled trials (RCTs) and long-term studies are needed to evaluate efficacy, side effects and individual response to the diet, it is imperative to study and understand the metabolic profiles of patients with epilepsy in order to isolate which dietary restrictions are necessary to maximize clinical benefit.

## 1. Introduction

Epilepsy in the pediatric and adolescent population is a devastating condition where individuals are prone to recurrent epileptic seizures or a change in behavior or movement that is the direct result of a primary change in the electrical activity in the brain [1]. Up to 65% of individuals with epilepsy will have seizures controlled with antiepileptic drugs (ASMs) or enter spontaneous remission in their lifetime [2]. However, this leaves a large percentage of patients that are refractory to drug therapy. Current methods for treating refractory epilepsy include surgery, vagus nerve stimulation or a Ketogenic Diet (KD).

Prior to development of the KD, historical iterations of the use of fasting to treat epilepsy have been documented from at least 500 BC. Fasting was noted as a therapeutic measure against epilepsy in the Hippocratic collection and was referred to again in the King James Version of the Bible 5 centuries later [3]. In 1921, Woodyatt et al. observed that ketones, specifically acetone and β-hydroxybutyric acid, appeared in subjects either by starvation or by consuming a high fat and very low carbohydrate diet [4]. In the same year a case series of a ketone-producing diet in patients with epilepsy was published by a Mayo Clinic physician [5] with the KD subsequently introduced in 1924 after 2 years of clinical trial [6]. The diet was widely used as a treatment for epilepsy in the 1920s and 1930s but its use was significantly decreased with the discovery of new and more effective ASMs [3]. These ASMs had fewer side effects than those previously available and were much easier to adhere to than a restrictive diet. The KD regained popularity in the 1990s after promising research was published and the Charlie Foundation, a non-profit which provides information about diet therapies for people with epilepsy, was created after a 2 year old child was cured of his epilepsy after being on the therapy [7].

### Overview of Ketone Body Metabolism

Ketogenesis or the production of ketone bodies (KBs) primarily occurs in liver from fatty acid β-oxidation-derived acetyl-CoA and transported to the extrahepatic tissues for terminal oxidation [8]. These KBs are involved in a variety of important metabolic pathways such as fatty acid β-oxidation (FAO), gluconeogenesis, the tricarboxylic acid (TCA) cycle, de novo lipogenesis and sterol biosynthesis [9,10]. This metabolic mechanism provides an alternative source of energy, especially under a fasting state, during which the availability of carbohydrate is limited while the availability of fatty acids increases, therefore serving as the main energy source [10,11]. Ketone bodies, β-hydroxybutyrate (BHB) in particular, have been traditionally considered an alternative source of energy supply [9,12,13] and metabolism in humans provides a significant source of fuel for the brain in a carbohydrate-limited state [14]. Brain cells are able to produce energy from glucose and ketones and are therefore considered metabolically flexible. [15]. During periods of very low carbohydrate intake, BHB is the primary energy source for neurons [16].

## 2. Materials and Methods

A literature search was performed using MEDLINE and PUBMED to locate peer-reviewed articles of observational studies, clinical trials or meta-analysis reporting the results of ketogenic diet therapy on refractory epilepsy outcomes in children and adolescents. Articles were reviewed for inclusion based on their relevance to this subject matter. Conference papers or reports and case studies were not included.

## 3. Results

### 3.1. Ketogenic Diets

The term Ketogenic Diet or Ketogenic Diet Therapy (KDT) refers to any diet therapy in which dietary composition results in a ketogenic state of human metabolism [17]. The diet generally refers to a high fat, low carbohydrate and moderate protein diet. Following the development of the classic ketogenic diet (cKD), new diets have been proposed in an attempt to increase retention and palatability while mimicking the effects produced by the original diet. Currently, there are 4 major KDTs (Table 1): the cKD, the modified Atkins diet (MAD), the medium chain triglyceride ketogenic diet (MCTKD) and the low glycemic index treatment (LGIT) [18].

### 3.2. Classic Ketogenic Diet

The cKD as proposed by Wilder et al. in 1921 is most often based upon consumption of a large amount of long-chain saturated triglycerides (LCTs) and small amounts of medium chain triglycerides (MCTs) in a 3:1 to 4:1 ketogenic diet ratio (KD ratio) of fat to carbohydrate and protein [5,19]. It is the most restrictive of the described KDTs. Implementation of the KD varies to but was been historically initiated in a hospital after a fast of 2 to 3 days or until ketone bodies are produced by the liver. Present day, initiating the cKD with a fast is considered an antiquated practice and many centers do not require a fast whatsoever. Achievement of a ketotic state is followed by a gradual introduction of the cKD over a 3-day period, working up to a KD ratio of 3:1 to 4:1 or until adequate ketosis is reached, generally a minimum of 80–160 mmol/L BHB. All foods and beverages must be precisely weighed on a gram scale, as is the common practice for many KDT initiations and calorie and protein restrictions may be recommended [20,21]. Starting the cKD in an inpatient setting not only allows the child to be closely monitored but also allows ample time to train caregivers. Subtypes of the cKD have recently increased in popularity. Subtypes continue to follow similar macronutrient distributions but alters macronutrient sources. The Mediterranean KD was developed to incorporate the health benefits of the Mediterranean diet which has been shown to lower rates of coronary heart disease, hypercholesterolemia, hypertension, diabetes and obesity [22]. The Mediterranean KD is an unlimited calorie diet focused on the integration of virgin olive oil as the main fat source fish as protein [23]. Components of this diet would likely greatly increase intakes of polyunsaturated and monounsaturated fatty acids along with dietary antioxidants. Another diet, the carnivore diet, encourages an animal-based meal pattern by eliminating all foods but meats, fish and eggs. The carnivore diet is characterized by high intakes of saturated fatty acids and medium to long chain triglycerides [24]. While there are limited data currently available on cKD subtypes, these variations may warrant exploration in patients with epilepsy in the future.

#### 3.2.1. Classic KD Efficacy

Initial cohorts and randomized controlled trials (RCTs) validated the ability of the cKD to significantly reduce seizure frequency [25,26] while later studies attempted to assess the need for high KD ratios [27,28,29,30]. A prospective cohort of children on a cKD with a 2:1 KD ratio found a greater than 50% seizure reduction was achieved in 52%, 43%, 40%, 33% and 30% of subjects at 3, 6, 12, 24 and greater than 3 years follow up. [30]. An RCT published in 2011 aimed to assess the efficacy of a higher versus a more liberal KD ratio reported a >50% reduction in seizures at 3 months for 58% of patients in the 4:1 group and 63% in the 2.5:1 group [27]. Although these data are not significant (*p* = 0.78), the results indicate that a 2.5:1 KD ratio may be as effective as the 4:1 KD. However, these results may not be generalizable as a study with a high number of patients with Lennox-Gastaut syndrome had better seizure control on a 4:1, rather than a 3:1 KD ratio (*p* < 0.05) [28]. Evidence for short-term efficacy has been reported with long-term efficacy appearing promising. In a 2018 review, seizure freedom on a 4:1 KD ratio diet was found to be as high as 55% after 3 months with seizure reduction rates at 85% [29]. These results are reflected in meta-analyses where at least 50% of children having at least a 50% reduction in seizures at 6 months [31,32]. In 2011, children from the original 150 patient cohort published in 1998 were contacted 3 to 6 years later to assess efficacy past 2 years [33]. 65 (43%) of subjects had a greater than 50% seizure reduction, including many who were now off the cKD. Of the children who remained on the diet for more than 3 years, 13 of 15 (85%) had greater than 50% reduction in seizures. Wiljnen et al. reported 35% of KD participants had a seizure reduction of greater than 50% from baseline compared to 18% of the control group at 16 months [34]. At Johns Hopkins Hospital, the cKD was shown to be effective beyond 6 years in 28 of 600 original patients that started the diet in 1994 [35]. Children were able to maintain large urinary ketosis and control was stable, with only periodic break-through seizures. Of the 28 children, 24 (86%) were reported to have a greater than 90% seizure reduction. Long-term cKD efficacy is often reported as ability for patients to achieve seizure freedom. Meta-analyses of early studies and case series have found rates of seizure freedom from 15%–20% [31,36,37]. A retrospective cohort of 276 children started on a cKD found that 65 patients (24%) became seizure free for a minimum of 1 month during treatment [38]. Of the patients that were initially seizure free in the cohort, 82% had seizure recurrence with only a 3% likelihood of remaining completely seizure-free at 18 months. Authors note seizure severity is so great for many children starting the cKD that meaningful improvement is the goal of the therapy, not necessarily seizure freedom. Therefore, the probable occurrence of a breakthrough seizure does not necessarily indicate the start of a decline back to baseline seizure frequency.

#### 3.2.2. Classic KD Efficacy in Infants

In the past, KDTs were not recommended for infants because infancy is a critical period for neurodevelopment and due to the perceived risk of inadequate nutrition [39]. Compared with adults, neonates and infants have the advantage of having a metabolism associated with the use of ketone bodies [40,41,42]. The efficacy of dietary treatment in infants is likely due to the higher amounts of enzymes that metabolize ketones and the production of monocarboxylic acid transporters, leading to more ketone bodies that can cross the blood-brain barrier (BBB) [40,43].

Nordli et al. published one of the first reports showing that KD could be safe and effective in a small group of infants [44]. Clinical practice shows the cKD with a 3:1 ratio is routinely used in infants in order to meet protein requirements with several studies prescribing a KD ratio of 2.5–4.1 [45]. However, the consensus statement published in 2016 recommends all young infants should be started on a 1:1 KD ratio then to adjust the ratio based on the level of ketosis and tolerance [46]. A study by Wirrell et al. in 2018 evaluated the tolerability and efficacy of the cKD in infants less than 12 months of age [47]. The diet was initiated in the hospital in all but 1 infant with a 2:1 ratio. Infants were not required to fast prior to initiation and were started on full calories to decrease the risk of hypoglycemia. Responder rates at 1, 6 and 12 months were 68%, 82% and 91%, respectively with 20%, 29% and 27% achieving seizure freedom. Infants who were underweight at diet onset showed increases in z scores of infant weight over time. However, those who were overweight at diet onset tended to show a mild increase in the z score short term but those remaining on the diet showed a decrease in the z score over the longer term. Currently, KDTs have been shown to be safe and effective for infants as young as 6 weeks [48] with some evidence suggesting that children younger than 2 years of age may be an ideal age population to start the cKD [45,49]. In fact, Le Pichon et al. studied a cohort of 9 infants between the ages of 1 and 13 months with the KD while maintained on breast milk [50]. All 9 infants achieved and maintained ketosis effectively.

### 3.3. Medium Chain Triglyceride Diet

The medium-chain triglyceride MCTKD was introduced by Huttenlocher in 1971 as the first option as an alternative to the cKD [51]. While cKD, is predominantly composed of LCTs the MCTKD encourages MCTs as large percentage of daily fat intake [18]. Commercial MCT oils, generally isolated from a 50:50 of coconut or palm oil, yield more ketones per kilocalorie of energy than LCTs and do not require carnitine to breakdown the fat source and are carried directly from the digestive system to the liver for ketone production by the portal vein [26]. MCTs are also able to be oxidized by the brain after crossing the BBB to provide a direct brain fuel source [52]. This increased ketogenic potential means less total fat is needed in the MCTKD, allowing for increased carbohydrate and protein intake and likely resulting in more food choices. In the traditional MCTKD, MCTs provide 60% of the diet’s energy which can lead to adverse gastrointestinal effects such as vomiting, diarrhea and abdominal pain [51]. Therefore, a modified MCTKD, using 30% of energy from MCTs and 30% from LCTs, was developed [53]. This diet is generally implemented in a hospital setting and may require a fast and a calorie restriction, much like the cKD.

#### Medium Chain Triglyceride Diet Efficacy

Schwartz et al. initially compared the KD (4:1 KD ratio) with the traditional MCTKD and the modified MCTKD diet in 55 children and 3 adults to assess efficacy [54]. In the non-randomized study, seizures were effectively controlled equally by all 3 diets in children less than 15 after 3 weeks. Neal et al. [55] was able to assess efficacy of these diets at 3, 6 and 12 months, finding no statistically significant differences in reported mean percentage of baseline seizures among the MCTKD and cKD groups (3 months: cKDl 66.5%, MCTKD 68.9%; 6 months: cKD 48.5%, MCTKD 67.6%; 12 months: cKD 40.8%, MCTKD 53.2%; all p > 0.05). Serum acetoacetate and BHB levels at 3 and 6 months were significantly higher in children on the cKD (*p* < 0.01). This should be expected as the MCTKD allows more carbohydrates. Correlations between acetoacetate and BHB levels and seizure control were significant for both ketone bodies at 3 months (acetoacetate, *p* < 0.009; βHB, *p* < 0.036) but not at 6 and 12 months. In a more recent study, of 16 Thai children started on the MCTKD, 64.3% reported >50% seizure reduction by month 3 with 27.6% becoming seizure free. [56]. Although multiple studies report overwhelmingly positive results in reduction of seizure frequency [54,55,57,58], even in super refractory status epilepticus [59], a systematic review of the effectiveness of a cKD with refractory epilepsy showed limited evidence that the it is more effective in reducing the frequency of seizures compared to the MCTKD [58].

### 3.4. Modified Atkins Diet

The MAD was created at Johns Hopkins Hospital in 2003 as an attempt to create a more palatable and less restrictive dietary treatment, primarily for children with behavioral difficulties and adolescents that parents and neurologists were reluctant to start on the cKD [60]. While starting the cKD often requires fasting and an inpatient stay of 1 to 3 days, the MAD is commenced on an outpatient basis with no need for hospitalization [61]. The MAD is considered less restrictive alternative to the cKD as protein intake is not limited and calories are never restricted. [60]. Foods and food portions are estimated, therefore eliminating the need to weigh food on a scale. The KD ratio of MAD is 1:1 to 2:1. For every 1 to 2 g of fat there is 1 g of carbohydrate plus protein [62]. This diet is beneficial for patients who have difficulty tolerating the restrictiveness of the cKD or those with a history of response to cKD but inability to maintain due to restrictiveness. The MAD is also an excellent option for adolescents and adults with a larger appetite, needing quick dietary intervention on out-patient basis and limited time or resources for the cKD [60]. Upon implementation, carbohydrates are limited to 10 g per day in children. Carbohydrates are increased after 1 month to 15 g, then increased further to 20 to 30 g per day as tolerated, based on seizure activity. In contrast to the LGIT, all carbohydrates irrespective of glycemic index (GI) are allowed [21]. The diet is “modified” from the colloquially popular Atkins diet as the “induction phase” of the diet, which limits carbohydrates, is maintained throughout diet therapy, intake of fat is encouraged and the goal is for seizure control, not weight.

#### Modified Atkins Diet Efficacy

Several studies have shown the efficacy of the MAD since its development in 2003 with studies reporting at least 30% of study subjects having a > 50% reduction in seizures in the study period. [63,64,65,66]. Unfortunately, even with an increased allowance of carbohydrates, families found the diet very restrictive and had difficulty finding recipes suitable for the children. Additionally, Kossoff et al. [63] reported quality of life increased from 43% to 68% in responders but decreased from 49% to 39% in non-responders.

Sharma et al. [66] published the first open label, non-blinded RCT on the MAD in 2013. Intake of carbohydrate in the intervention group was restricted to 10 g per day. At 3 months, mean percentage of baseline seizure frequency at 3 months was 59% (95% CI 44–74.5) in the intervention group as compared to 95.5% (95% CI 82–109) in the control group. The proportion of children with a greater than 90% seizure reduction (30% vs 7.7%, *p* = 0.005) and a greater than 50% seizure reduction (52% vs. 11.5%, *p* < 0.001) was significantly higher in the diet group.

Kim et al. studied the efficacy of the cKD and MAD in refractory childhood epilepsy. Subjects in the cKD group were prescribed a 4:1 diet with non-fasting initiation protocol while those in the MAD group were restricted to 10 g of carbohydrate per day and permitted to increase by only 5 g per day up to 10% carbohydrate by weight. However, calories were restricted to 75% recommended daily intake. Authors reported that over-all efficacy of the 2 diets was comparable except for in subjects aged 1 to 2 years-old. At the 3-month assessment, those in the KD group had a baseline seizure frequency of a 38.6% while the MAD group’s was 47.9%. After 6 months, baseline percentage was 33.8% in the cKD intervention group and 44.6% in the MAD group. The cKD group had lower baseline seizure percentages compared to the MAD group at 3 and 6 months but the differences were not statistically significant (95% CI: 24.1–50.8; *p* = 0.291 at 3 months, 95% CI: 17.8–46.1, *p* = 0.255 at 6 months). When analyzed by age group, the mean and median baseline seizure frequency in participants less than 2 years were much lower in children on a cKD compared with those on the MAD. 3 months after starting diet therapy, 9 participants following the cKD had seizure freedom while only 3 achieved this on the MAD (*p* = 0.047). With these results, authors posit the KD may not have a definite advantage over the MAD in subjects less than 2 years old. A systematic review and meta-analysis compared short-term and long-term efficacy of MAD vs cKD [67]. Pooled efficacy rate for greater than or equal to 50% seizure reduction of the cKD at month 3, 6 and 12 was 53% (95% CI: 42–63), 46% (95% CI: 36–56) and 41% (95% CI: 27–55), respectively. Efficacy of the MAD for greater than or equal to 50% seizure reduction was 52% (95% CI: 41–62) at month 3, 45% (95% CI: 34–56) at month 6. Studies comparing the cKD and MAD for greater than or equal to 50% seizure reduction at month 3 and month 6 found no significant difference (*p* = 0.367). This result was echoed when assessing a greater than 90% seizure reduction of the diets with no statistically significant difference (*p* = 0.235).

### 3.5. Low Glycemic Index Treatment

The term GI describes the tendency of foods to elevate blood glucose [68]. GI is calculated from the incremental area under the blood glucose curve after feeding, indexed to ingested glucose which equals 100. Foods with high GI (e.g., most refined carbohydrates) produce substantial increases in blood glucose and insulin levels, whereas foods with a low GI (e.g., meat, dairy, some fruits, some vegetables and some unprocessed whole-grain foods) induce lower postprandial plasma glucose and insulin profiles [69]. Abrupt changes in blood glucose levels have been previously shown to reduce seizure threshold in mouse models [70].

The LGIT was initially introduced as a liberalized version of a KDT and was first found to be successful in Massachusetts General Hospital in 2005 [21]. The diet is more liberal and allows 40 to 60 g carbohydrate per day but restricts sources of carbohydrates to a GI less than 50 to prevent postprandial increases in blood glucose. Fats and proteins are unrestricted. This diet is preferable in adolescents as KD may have significant compliance issues. Implementation of treatment does not require admission to hospital.

#### Low Glycemic Index Treatment Efficacy

With the relatively recent development of the LGIT, fewer data, specifically on long-term efficacy, are available for review. However, published cohorts consistently report decreased seizure frequency for subjects following the therapy [71,72,73]. A retrospective chart review of 131 children initiating the LGIT was published by Muzkewicz et al. in 2009 [72]. The LGIT was initiated in an outpatient setting with follow up visits every 1 to 3 months. Carbohydrate intake while on the LGIT ranged from 15 to 150 g per day. At 3 months, mean carbohydrate intake was estimated at 53 plus or minus 18 g per day. Seizure reductions of greater than 50% from baseline at 1, 3, 6, 9 and 12 months measured 42%, 50%, 54%, 64% and 66%, respectively at the follow up intervals. Efficacy did not differ significantly with regard to seizure type (*p* > 0.05).

Continued efficacy was reported in a cohort of 42 pediatric patients who were educated by a dietitian regarding the restriction of carbohydrates with high glycemic index (GI > 50) and limiting total carbohydrates to 40–60 g per day (roughly 10% of daily calories) [71]. Mean seizure frequencies at 2, 3 and 8 weeks after treatment were lower than baseline with seizure reduction rates of 56%, 61% and 67%, respectively (*p* < 0.001). Serum glucose was significantly lower after LGIT was introduced (*p* = 0.005) but importantly, was not found to be correlated with seizure control.

The DIET-Trial published in 2018 randomly assigned 167 children to receive the cKD, MAD or LGIT [74]. After 24 weeks, mean percentage seizure reduction (±SD) was −60.3 (±32.8) in the KD sub-group; −47.9 (±45.9) in MAD sub-group; and −54.7 (±40) in LGIT sub-group. Mean difference in seizure reduction between the cKD and LGIT interventions was −5.66 (95% CI: −17.34, 6.02) indicating the LGIT is not inferior to the MAD and cKD treatments.

Of the multiple studies comparing KDTs, no KDT proved to be superior (Table 2) [49,53,54,55,67,74]. In fact, Wibisono et al. published a 10-year retrospective study of 48 children receiving either the cKD, MCTKD or MAD and found that diet duration or KDT did not predict reduction in seizures (*p* = 0.381; *p* = 0.272) [75]. It is important to note, however, that in many studies children on KDT remain on at least 1 other ASM.

### 3.6. Adverse Effects

Adverse effects of the various KDTs are reported in a large percentage of children and are often cited as a reason for participants dropping out of trials [29]. In a systematic review of cKD and MCTKD prospective studies, more than 40 categories of adverse effects were identified with gastrointestinal, cardiovascular, renal/genitourinary and skeletal being the most common [76]. Although side effects have been reported in all 4 KDTs, it appears stricter KDTs, such as the 4:1 cKD, are associated with higher incidences of adverse effects [29]. The frequency and severity of adverse effects from the different KDTs appear to increase with the restrictiveness of the diet. In fact, a 2018 RCT comparing the cKD with MAD and LGIT found significantly less adverse events in the LGIT intervention (*p* = 0.036) [74]. As previously discussed, the cKD generally requires a 1 to 3 day inpatient hospitalization, fasting upon initiation and a calorie restriction in addition to having the highest KD ratio [77].

#### 3.6.1. Gastrointestinal

Gastrointestinal side effects are the most frequently reported in children following a KDT. In a systematic review of published prospective studies, the most common individual side effect after consuming a cKD was constipation (175 cases, 13.2%) [76]. Constipation was also the most common side effect noted in the retrospective review by Lin et al. [78] and for both the cKD and MCTKD in the RCT by Neal et al. [55]. However, incidence of constipation was considerably higher at 3 months (cKD: 45%, MCTKD: 33%), 6 months (cKD: 48%, MCTKD: 41%) and 12 months (cKD: 45%, MCTKD: 39%), for those on the cKD. Constipation may be resolved by providing an enema, prescribing polyethylene glycol or by increasing dietary fibers [78,79]. However, others have found reducing or stopping fiber intake to be more effective in decreasing symptoms of constipation [80]. Further research into soluble versus insoluble fiber intake for constipation from KDT may be warranted to re.

Following constipation, side effects with the next highest incidence rates were gastrointestinal disturbances (9.6%) and vomiting (9.1%) [76]. In the RCT by Neal et al. the incidence of vomiting after 3 months for the cKD and MCTKD was similar between groups. [55]. However, vomiting was the most common side effect in the MCTKD while constipation was noted to be the most common in the KD. In 2016, Chomtho et al. notes nausea was a common adverse effect of the MCTKD in Thai children with intractable epilepsy in 2016 with 25% reporting the symptom [56]. Some suggest the MCTKD induces more vomiting and diarrhea in patients compared with the cKD [81]. Yet, the opposite was seen in the retrospective review published by Wibisono et al. where the cKD induced more vomiting [75]. It is important to distinguish if a study examining the MCTKD is using the classic (60% MCT) or modified (30% MCT) when assessing side effects. Treatment for nausea and vomiting on a KTD generally includes nausea medication, providing intravenous fluids or changing the KD ratio [78].

#### 3.6.2. Cardiovascular

Cardiovascular side effects, in particular high serum lipid profiles, are often suspected in children who follow a very high fat diet. Blood lipid levels were collected before and after 3 months of KD treatment in a study published in 2019 [82]. Triglyceride and total cholesterol levels were found to be slightly higher than those before the treatment and high-density lipoprotein (HDL) was slightly lower. However, differences were not statistically significant (*p* > 0.05). In patients following the MAD, Kossoff et al. reported mean triglycerides had a non-significant increase (*p* = 0.28, respectively) with a significant increase in cholesterol from 4.0 mmol/l (S.D. ± 0.9) to 5.3 mmol/L (S.D. ± 0.41) (*p* = 0.02). Conversely, in 2011, Miranda et al. found no significant increases in free cholesterol, triglycerides or LDL cholesterol after subjects followed an MAD for 12 months [64].

In a review by Cai et al., the incidence of dyslipidemia (hyperlipidemia, hypercholesterolemia and hypertriglyceridemia) was lower than that of gastrointestinal symptoms [76]. Hyperlipidemia had the highest incidence with 63 cases (4.6%) followed by hypercholesterolemia (53 cases, 3.8%) and hypertriglyceridemia (44 cases, 3.2%). 1 long-term study found mean cholesterol of subjects after 6 years on the diet was only slightly high at 201 mg/dL. HDL, low-density lipoprotein (LDL) and triglycerides were all within normal limits at 54 mg/dL, 129 mg/dL and 97 mg/dL, respectively [35]. It has been suggested that a standard lipid panel may not be sufficient to analyze the effect of KDTs on cardiovascular disease (CVD) [83]. Subfractions of HDL and LDL vary by particle size and density and have been proposed as a measure to improve assessment of CVD risk [84]. For example, large LDL particles may be, in fact, cardioprotective while small LDL particles contribute to plaque formation [85,86]. Further research on changes and concentrations of these subfractions while on KDT are needed to understand. When cardiac function was evaluated using imaging, researchers found no significant differences after 1 year of therapy [87]. Guzel et al. implemented the following interventions for those developing hyperlipidemia on an olive-oil based KD—modifying the KD to reduce dietary fat by 20%–25% without affecting blood ketone levels, eliminating egg yolk and saturated fat sources and providing atorvastatin (10 mg/day, by mouth) to block endogenous cholesterol biosynthesis [79]. Few data are available regarding long-term effects of the KDT in regards to cholesterol.

Aside from the concern of raising lipid profiles, the effect of KDT on QT intervals has been researched. A case report by Best et al. found 3 patients (15%) with a prolonged QT interval (QTc) after following the cKD [88]. Although multiple ASMs are known to contribute to prolonged QTc [89], authors in this case study posit a starvation-like state may create metabolic derangement conducive to cardiac conduction abnormalities and/or myocardial dysfunction. In fact, a case study of 2 children experiencing QT prolongation who died suddenly at home were found to exhibit selenium deficiency [90]. Magnesium depletion, likely from the low magnesium content of the KD, can also cause long QTcs and responds to magnesium replacement [89]. Increased magnesium content is one benefit of more liberal diets, like the MAD, which allow non-starch green vegetables Prospective studies on the controversial topic have been published finding the corrected QT interval did not change significantly over a 12-month period [91,92].

#### 3.6.3. Renal/Genitourinary

The cKD has been associated with a 3.1%–6.7% incidence of renal calculi [93,94]. In Freeman et al.’s prospective cohort, 3 of 150 children were found to have uric acid stones and 3 with calcium oxalate or calcium phosphate stones [95]. In 2000, a prospective cohort reported 6 of 112 patients initiating the cKD developed kidney stones during the follow-up period of 2 months to 2.5 years [96]. In the long term follow up of patients on the cKD for greater than 6 years, kidney stones occurred in 7 children, suggesting the incidence of renal calculi may increase with longer term use of the diet [35].

It is recommended that patients commencing the cKD are prescribed potassium citrate (2 mmol potassium/kg/day) for the duration of treatment. to prevent renal calculi [97,98]. Potassium citrate counteracts acidosis and increased bone demineralization by alkalinizing urine and solubilizing free calcium. McNally et al. reported the use of potassium citrate led to a 7-fold reduction in renal calculi in patients following the cKD [99]. Potassium citrate supplementation is severely constipated and steps may need to be taken to mitigate this unfortunate side effect.

#### 3.6.4. Skeletal

Regarding bone health, many early studies did not find any adverse effects of the cKD on bone health [100,101]. However, these studies were limited by small sample sizes and short-term follow up. In 2008, Bergqvist et al. found whole body and lumbar spine bone mineral content for age declined by 0.6 standard deviations per year [102]. The next year, the 2009 International Ketogenic Diet Study group consensus-based guidelines recommended incorporating dual energy X-ray absorptiometry (DEXA) scans periodically for assessment of bone health [103]. Simm et al. aimed to assess the effect of a cKD on bone health outcomes and found a trend towards reduction in lumbar spine bone mineral density with an average decrease of 0.16 standard deviations (relative to age-matched referent children) for every year a subject remained on the diet. [104]. These results reflect those by Bergqvist in 2008.

On the other hand, in a cohort of patients with GLUT1 deficiency, no adverse effects in bone composition and mineral content were detected after 5 years in ketosis with vitamin supplementation. Again in 2019, Svedlund et al. found no negative effect on bone mass in children following the MAD. Although there are also reports of no adverse effects in bone composition or mineral content in other studies [105,106], future human studies evaluating KDTs on bone health as well as potential interventions to prevent the decline in bone mineral density are needed.

#### 3.6.5. Growth

Etiology of poor growth in children on KDTs include inadequate energy and protein intake, the underlying condition and treatments and acidosis or ketosis [107]. Systematic reviews have found inconsistent results on the KD’s effect on growth with some showing a positive impact while others show a negative impact [76].

Williams et al. conducted a retrospective cohort of 21 children following a cKD and reported 86% of the children’s height percentiles fell from their initial visit (*p* = 0.001) with statistical analysis supporting the hypothesis that growth for children on the diet is different from normal growth [106]. Another cohort of 237 children on a cKD were reported to have a slight decrease in height Z-scores during the first 6 months on a cKD with more substantial changes at 2 years [97]. However, in both of these studies, the cKD was characterized by a 75% to 85% calorie restriction to achieve ketosis [18]. In 2005, Peterson et al. reported height Z-score significantly decreased between 6 and 12 months on cKD with marked ketosis but not in subjects with moderate ketosis [98]. Spulber et al. also reported a negative correlation between growth rate and BHB concentrations (Pearson *r* = −0.48; *p* < 0.05) [108]. Again, the target of the total daily caloric intake was based on approximately 75% of the recommended daily allowances for the child’s desirable weight for height.

Other studies, specifically those that do not require a calorie restriction and provide vitamin and mineral supplementation, do not report such significant impacts on growth [63,107]. A prospective cohort of 151 children with refractory epilepsy were followed to evaluate growth as well as nutritional status on the cKD [109]. Participants in this cohort were not calorie restricted, received 90% of energy from fat and 10% of energy from protein plus carbohydrate and a sugar-free multivitamin, calcium and potassium citrate as oral supplementation. Patients were monitored weekly during the first month and every 3 months following. 45 patients remained on the cKD for 24 months. During this time, only 3 patients were found to have linear growth decreased by more than 1 standard deviation. 

### 3.7. Ketogenic Diet Mechanism

To date, the exact etiology of epilepsy remains unknown, as does the mechanism of the KD. the KD imitates the biochemical effects of fasting and induces a shift away from glycolic energy production toward energy generation through oxidative phosphorylation leading to fatty acid β-oxidation and ketone production [19]. BHB is the predominate ketone measured in the plasma or urine and has been used as a clinical measure of KD implementation. Although significant elevations of BHB are seen during KDT, there are currently no significant correlations between levels and seizure activity [110]. Epilepsy is characterized by abnormal brain activity thought to be due to factors to include genetics, developmental disorders or neurologic damage such as a traumatic brain injury, hypoxia or fevers [111]. These factors, in turn, may lead to hyperexcitability and seizure activity by negatively impacting neurotransmitters, ion channels, neuroinflammation, oxidative stress or the microbiome. Many hypothesize the KD mechanism of action includes halting or reversing one or more of these negative outcomes.

Ketosis may protect against seizures by altering ion and neuronal activity [112,113]. Glutamate and γ-aminobutyric acid (GABA) are chief excitatory and inhibitory neurotransmitters, respectively [114] One mechanism through which ketone bodies may lead to seizure reduction could be through more efficient glutamate recycling [115,116]. Metabolism of acetyl-CoA produced from a high-fat diet requires the use of oxaloacetate in the TCA which correlates with increased availability of α-ketoglutarate, low aspartate levels and high glutamate levels. This may increase GABA synthesis, limit reactive oxygen species (ROS) generation and boost energy production. Elevated GABA levels have also been shown to stimulate chloride channel receptors which causes an increase of negatively charged ions and induces hyperpolarization thereby inhibiting activities required for neuronal excitation. Seizure presence may also be affected by other neurotransmitters, including norepinephrine but current research remains inconclusive [117,118,119].

Another hypothesized mechanism of action includes mitochondrial function and the production of reactive oxygen species (ROS). Animal models have shown mice and rats being fed a KD had higher levels of uncoupling proteins in the hippocampus, more mitochondria and fewer ROS [120,121]. Although the KD’s impact on mitochondrial function remains unknown, ROS production has been shown to decrease with the presence of ketone bodies [122]. Glutathione, an intercellular antioxidant able to prevent damage caused by ROS, also appears to increase when BHB or acetoacetate are present [123]. Increased glutathione and reduced ROS can protect neurons and inhibit cell death [124].

Brain damage leads to a response of pro-inflammatory molecules such as cytokines and chemokines which can lead to epilepsy onset or potentiate recurrent seizure development through altered neural connectivity, a hyper-excitable neuronal network and extra-hippocampal neuronal cell death and gliosis [125,126,127]. Furthermore, the increase in neuroinflammation can result in the breakdown of the BBB [128]. Multiple hypothesis regarding the effect of ketosis on inflammation have been researched in recent years. Youm et al. [129] has shown BHB but not acetoacetate, butyrate or acetate, is able to inhibit nucleotide-binding domain (NOD)-like receptor protein 3 (NLRP3) inflammasome production. NLRP3 inflammasomes are hallmarks of multiple autoinflammatory syndromes and are involved in the pathogenesis of metabolic disorders [130]. Neuroprotective and anti-inflammatory effects of ketosis could also be due to BHB’s ability to activate the hydroxycarboxylic acid 2 (HCA2) receptor which causes Ly-6C^Lo^ monocytes and macrophages to send a neuroprotective signal to the brain [131].

Microbiomes are important aspects of brain activity and behavior as well as immune function [132]. The concept of a gut-brain axis was developed from studies correlating changes in the microbiome to neurological disorders [133]. The gut-brain axis refers to the bidirectional communication between the central nervous system and the enteric nervous system which links brain and intestinal function [134]. Ketosis may mediate the immune response seen in KDTs by altering the gut microbiota [135]. Intake of a KD has been shown to change microbial structure and function as well as decrease levels of *Actinobacteria* and *Bifidobacterium* while increasing *Proteobacteria.* The decrease in levels of *Bifidobacterium* inhibits pro-inflammatory Th17 cells in the intestine [136,137]. KDT has also been connected with the increase of *Akkermansia Muciniphila* and *Lactobacillus*, which generate short chain fatty acids (SCFAs) such as butyrate, acetate, propionate and valproic acid [138]. SCFAs are thought to play an important role in the gut brain axis by crossing the BBB and maintaining its integrity [139].

Although it is understood MCTs in the MCTKD provide increased ketogenic potential and direct metabolism, mechanism of action remains unclear. MCTs are mainly composed of the unbranched fatty acids octanoic (C8:0) and decanoic (C10:0) acids which produce ketones [140]. Recent data reports conflicting results on seizure control potential of these medium chain fatty acid intermediates [141,142,143]. Brain cells treated with C8:0 produce larger amounts of ketone bodies and butyrate than C10:0 and application prior to seizure induction has been shown to increase seizure threshold [143] However, increased glutathione concentrations have been seen with cells incubated with C10:0 and increased seizure control has also been seen with C10:0 in an in vitro seizure model [141,142]. Further research on medium chain fatty acid intermediates is needed to investigate the potential anti-seizure effect in humans.

## 4. Conclusions

Since the 1930s, observational trials, RCTs and meta-analyses have consistently shown KDTs are able to significantly reduce seizure frequency in children and adolescents with refractory epilepsy. However, the evolution from the most restrictive KDT, the cKD with a 4:1 KD ratio, to more liberal MAD and LGIT options with increased carbohydrate and protein provision, do not appear to impact efficacy. In fact, no KDT has shown superiority in an RCT in patients 2 to 18 years of age. As shown in Table 1, each KDT recommends carbohydrate intake to be less than 20% of an individual’s total daily intake percentage, suggesting that a change in metabolism contributes to the reduction in seizure frequency. However, we still do not understand the true impact of a restrictive diet in children and adolescents on health outcomes, specifically those experiencing significant adverse effects. While many short-term side-effects (i.e., nausea, vomiting), have relatively straightforward interventions, long-term side-effects such as bone health and hypercholesterolemia require further investigation. Ultimately, the adequate growth and cardiac function of children on restrictive diets is of utmost concern. Close monitoring by an interdisciplinary team is imperative to ensure the benefits of KDT outweigh any potential risks. Early identification and discontinuation of KDT in children and adolescents not benefiting from therapy could decrease the impact of adverse effects in this vulnerable population.

Combined analysis of KDT studies is very difficult secondary to variations in population characteristics, diet prescriptions, observation periods and measured parameters among the studies. Although macronutrient distributions are similar among the KDTs, data regarding fatty acid compositions of the diets and dietary antioxidant intake is limited. As current research has shown BHB levels do not correlate with seizure frequency, we may instead need to identify which types of fats or fatty acid intermediates elicit anti-seizure effects. Currently available data is likely confounded by the heterogeneity of the diets and the etiology of the seizures themselves. Additionally, with evidence indicating neuroinflammation and presence of ROS act as a pathogen of epilepsy, intake of antioxidants from the diet could impact seizure frequency. The evaluation of long-term efficacy and side effects is particularly difficult due to the inability of the subjects to continue a KDT over an extended period of time. Retention rates on the cKD at 1 year and 2 years have been reported at 45.7% and 29.2%, respectively with the majority of subjects discontinuing KDT due to lack of efficacy [76,144]. These individuals are often classified as non-responders. Attrition rate is also largely affected by poor educational and support systems in place for patients and families. Although the retention rate for individuals on KDT remain low, it is important to note that there are reports of individuals staying on the diet for a prolonged period of time [32,145]. We hypothesize individuals who remain on KDT are often responders (patients with ≥ 50% seizure reduction) and likely continue therapy due to the reduction in seizure frequency and improved quality of life.

This review suggests that with the current paucity of evidence, there is not a clearly superior diet therapy nor a clear understanding of long-term health outcomes. Therefore, in regards to efficacy and safety, providers are unable to confidently prescribe one diet one over another. The ability to provide safe and effective nutrition intervention to children and adolescents with refractory epilepsy is of utmost importance. An important step to developing appropriate nutrition interventions may be to assess why KDTs are effective in some individuals but not others. There is a well-established fact that there are both responders and non-responders to KDT. However, with the exception of specific metabolic disorders, predictors of the response to therapy is unknown [146,147]. It is entirely possible that some patients might respond better to diets that induce higher BHB levels while others may require a C8:0 MCT to control seizures. Comprehensive metabolic profiling may be a useful tool to identify metabolic signatures of seizures. Metabolic profiling or “metabolomics,” is the analysis of the all measurable metabolites under a given set of conditions [148]. Recently, Boguszewicz et al. utilized metabolomics to detect children with epilepsy had increased levels of serum N-acetyl-glycoproteins, lactate, creatine, glycine and lipids [149]. These metabolites are involved in multiple metabolic pathways including the TCA cycle, glutathione metabolism and several amino acid metabolisms. The ability to identifying the metabolic signature of seizures could allow practitioners to develop targeted and individualized nutrition therapies in the future. We posit the provision of targeted nutrition interventions will often times be less restrictive and more effective, therefore decreasing adverse effects and increasing the prevalence of KDTs responders. Ultimately, our goal is the ability to provide a specific dietary treatment based on the patient’s metabolic signature to decrease seizure frequency and improve the quality of life for children with refractory epilepsy.

## Figures and Tables

**Table 1 nutrients-12-01809-t001:** Macronutrient Distributions across KDTs.

KDT	Clinical Implementation	Diet Pattern	Percent Total Daily Energy Intake		
			Fat	Carbohydrate	Protein
KD	Inpatient Stay	KD Ratio of 3:1–4:1	90	4	6
Traditional MCTKD	Inpatient Stay	60% total energy intake from MCT	70–75	15–18	10
Modified MCTKD	Inpatient Stay	30% total energy intake from MCT, 30% from LCT	70–75	15–18	10
MAD	Outpatient	KD ratio of 1:1–2:1	60–65	5–10	30
LGIT	Outpatient	40–60 g carbohydrate per day Restricts carbohydrate sources to a GI < 50	60	10	30

KDT: Ketogenic Diet or Ketogenic Diet Therapy, cKD: the classic ketogenic diet, MAD: the modified Atkins diet, MCTKD: the medium chain triglyceride ketogenic diet, LGIT: the low glycemic index treatment.

**Table 2 nutrients-12-01809-t002:** Efficacy Comparisons of KDTs.

Author	Study Design	Diets Studied	Conclusion
Schwartz et al. [53]	Cohort	1. cKD 2. Classic MCT 3. Modified MCT	All 3 KDT were shown to be effective in short-term management of children
Neal et al. [55]	RCT	1. cKD 2. MCT	cKD and MCTKD protocols are comparable in efficacy and tolerability
Kim et al. [49]	RCT	1. cKD 2. MAD	The MAD might be considered the primary choice for the treatment of intractable epilepsy in children but the cKD is more suitable for first-line therapy in patients < 2
Rezaei et al. [67]	Systematic Review and Meta-Analysis	1. cKD 2. MAD	cKD does not differ substantially from MAD in ≥ 50% and ≥ 90% reduction in seizure frequency at 3 and 6 months
Sondhi et al. [74]	RCT	1. cKD 2. MAD 3. LGIT	LGIT is not inferior to MAD or cKD KDTs
Wibisono et al. [75]	Retrospective Cohort	1. cKD 2. MCTKD 3. MAD	3 KDTs were comparably effective in seizure control

KDT: Ketogenic Diet or Ketogenic Diet Therapy, cKD: the classic ketogenic diet, MAD: the modified Atkins diet, MCTKD: the medium chain triglyceride ketogenic diet, LGIT: the low glycemic index treatment.

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
