# Peer review of "Efficacy and Safety of a Ketogenic Diet in Children and Adolescents with Refractory Epilepsy—A Review"

_nutrients, 2020, doi:10.3390/nu12061809_

Round 1

Reviewer 1 Report

The authors present a reasonably comprehensive, well-written review of the traditional and recently evolving formulations of ketogenic diets in the treatment of pediatric epilepsy.  Their detailed review of short- and long-term studies of efficacy in prospective cohorts is much appreciated. 

This manuscript would benefit from additional material addressing two topics:

In section 1.2 ‘Mechanism’, the expanding literature on the role of neural inflammation in  seizure etiology deserves a modicum of discussion [eg, ref by Rana et al, 2018].  This is highly appropriate to this topic due to the recent reports that: 

(1)  beta-hydroxybutyrate upregulates endogenous defenses against oxidative stress [ref by Schimazu et al, 2013] and blocks the NLRP3 inflammasome [ref by Youm et al, 2014]; and 

(2) a well-formulated ketogenic diet has been reported to have broad-spectrum and lasting anti-inflammatory effects  [refs by Forsythe et al, 2008; and Athinarayanan et al 2019].

This combination of observed increases in seizure prevalence in auto-immune inflammatory conditions (eg, rheumatoid arthritis and lupus), novel physiological mechanisms whereby BHB down-regulates oxidative stress and inflammation, and human studies demonstrating potent anti-inflammatory effects of nutritional ketosis would be a valuable addition to this topic. 

In the Adverse Effects section, lines 357-364, the association of long-QT syndrome and ketogenic diets is mentioned, but causality is tenuous at best.  Two additional contributing factors need to be mentioned. 

(1)  A number of anti-seizure drugs are known to contribute to prolonged QTc intervals [ref by Feldman & Gidal, 2013]. 

(2)  Magnesium depletion is a well-established cause of long QTc and torsades de pointes, and it responds to magnesium replacement [ref by. Hoshino et al, 2006].  Furthermore, the very restricted KDs rarely meet basic magnesium needs, particularly in growing children, and standard supplements are of questionable adequacy.   One benefit of expanded food options in the MAD may be the inclusion of non-starchy green vegetables containing appreciable magnesium. 

Specific comments

Line 58.  “They are considered…” replace with “They have been traditionally considered…”

Lines 75-78.  While a ketogenic diet by definition increases serum ketones, blood glucose is not usually decreased (and certainly not out of the normal range) unless starting at diabetes/pre-diabetes values. 

Line 111.  Please recheck your math.  One mM BHB is 104 mg/L.  This value as stated appears to be 10X that – i.e., on the threshold of DKA. 

Line 308.  Delete one of the two ‘The’. 

Line 311.  Perhaps this should be:  ‘1 to 3 day’.

Line 363.  Perhaps this should be:  ‘published finding that the corrected’.

Line 435.  Perhaps this should be:  ‘secondary to variations in population’.

Line 447.  Perhaps this should be:  ‘confidently prescribe one diet over another..’

Suggested references

Rana, A., Musto, A.E. The role of inflammation in the development of epilepsy.  J Neuroinflammation 15, 144 (2018). https://doi.org/10.1186/s12974-018-1192-7

Feldman AE, Gidal BE.  QTc prolongation by antiepileptic drugs and the risk of torsade de pointes in patients with epilepsy.  Epilepsy & Behavior.  2013; 26: 421-426

Hoshino K, Ogawa K,  Hishitani T,  Isobe T, Etoh Y.  Successful uses of magnesium sulfate for torsades de pointes in children with long QT syndrome.  2006; 48:112-117.

Shimazu,T., Hirschey, MD., Newman, J., et al. Suppression of Oxidative Stress by β-Hydroxybutyrate, an Endogenous Histone Deacetylase Inhibitor. Science. 2013;339(6116): 211–214

Youm, Y-H., Nguyen, KY., Grant, RW., et al. Ketone body β-hydroxybutyrate blocks the NLRP3 inflammasome-mediated inflammatory disease. Nat Med. 2015; 21(3): 263–269

Forsythe, CE., Phinney, SD., Fernandez, ML., et al. Lipids. 2008; 43:65–77 Comparison of Low Fat and Low Carbohydrate Diets on Circulating Fatty Acid Composition and Markers of Inflammation. Lipids. 2008; 43:65–77

Athinarayanan SJ, Adams RN, Hallberg SJ, McKenzie AL, Bhanpuri NL, Campbell WW, Volek JS, Phinney SD, McCarter JP.  Long-Term Effects of a Novel Continuous Remote Care Intervention Including Nutritional Ketosis for the Management of Type 2 Diabetes: A 2-Year Non-randomized Clinical Trial.  Front Endocrinol (Lausanne) 2019; 10: 348.

Author Response

Point 1:   The authors present a reasonably comprehensive, well-written review of the traditional and recently evolving formulations of ketogenic diets in the treatment of pediatric epilepsy.  Their detailed review of short- and long-term studies of efficacy in prospective cohorts is much appreciated. This manuscript would benefit from additional material addressing two topics:

In section 1.2 ‘Mechanism’, the expanding literature on the role of neural inflammation in  seizure etiology deserves a modicum of discussion [eg, ref by Rana et al, 2018].  This is highly appropriate to this topic due to the recent reports that:

(1)  beta-hydroxybutyrate upregulates endogenous defenses against oxidative stress [ref by Schimazu et al, 2013] and blocks the NLRP3 inflammasome [ref by Youm et al, 2014]; and

(2) a well-formulated ketogenic diet has been reported to have broad-spectrum and lasting anti-inflammatory effects [refs by Forsythe et al, 2008; and Athinarayanan et al 2019].

This combination of observed increases in seizure prevalence in auto-immune inflammatory conditions (eg, rheumatoid arthritis and lupus), novel physiological mechanisms whereby BHB down-regulates oxidative stress and inflammation, and human studies demonstrating potent anti-inflammatory effects of nutritional ketosis would be a valuable addition to this topic.

Response 1: Thank you for your comment.  Section 1.2 ‘Mechanism’ was greatly expanded upon and was moved to section 3.7, just above the conclusion. Thank you for including references which propelled this expansion and improved the scope of the review article.

Point 2:   In the Adverse Effects section, lines 357-364, the association of long-QT syndrome and ketogenic diets is mentioned, but causality is tenuous at best.  Two additional contributing factors need to be mentioned.

(1)  A number of anti-seizure drugs are known to contribute to prolonged QTc intervals [ref by Feldman & Gidal, 2013].

(2)  Magnesium depletion is a well-established cause of long QTc and torsades de pointes, and it responds to magnesium replacement [ref by. Hoshino et al, 2006].  Furthermore, the very restricted KDs rarely meet basic magnesium needs, particularly in growing children, and standard supplements are of questionable adequacy.   One benefit of expanded food options in the MAD may be the inclusion of non-starchy green vegetables containing appreciable magnesium.

Response 2: Thank you for sharing your expertise on this subject. The text has been updated to include these factors.

Point 3: “They are considered…” replace with “They have been traditionally considered…”

Response 3: The text has been corrected.

Point 4:   Lines 75-78.  While a ketogenic diet by definition increases serum ketones, blood glucose is not usually decreased (and certainly not out of the normal range) unless starting at diabetes/pre-diabetes values.

Response 4: Thank you for this observation.  Upon further review, this excerpt was removed from the text to promote clarity.

Point 5:   Line 111.  Please recheck your math.  One mM BHB is 104 mg/L.  This value as stated appears to be 10X that – i.e., on the threshold of DKA.

Response 5: Thank you for identifying this error.  The text has been updated to the correct measure.

Point 6:   Line 308.  Delete one of the two ‘The’.

Response 6: The text has been corrected.

Point 7:   Line 311.  Perhaps this should be:  ‘1 to 3 day’.

Response 7: Thank you for the observation, the text has been corrected.

Point 8:   Line 363.  Perhaps this should be:  ‘published finding that the corrected’.

Response 8: Thank you for the observation, the text has been corrected.

Point 9:   Line 435.  Perhaps this should be:  ‘secondary to variations in population’.

Response 9: Thank you for the suggestion, the text has been updated.

Point 10:   Line 447.  Perhaps this should be:  ‘confidently prescribe one diet over another.

Response 10: Thank you for the observation, the text has been corrected

Reviewer 2 Report

The authors provide a good overview of the literature comparing different ketogenic or pseudo-ketogenic interventions in pediatric epilepsy. My most pressing concern is the manuscript does not, in its current form, add to the state of knowledge. KD for epilepsy is a century-old and vetted intervention and many prior reviews have made similar points to those raised in this manuscript. Therefore, I suggest supplementation of the existing document with a more comprehensive discussion of the possible mechanisms (emphasis on the plural) that contribute to the anti-seizure effects of KDs. Such a hypothetical section would fit nicely before the conclusion and lead into the authors’ final well-made point that subcategorizing epilepsy cases based on metabolic signatures to provide personalized medicine is the future. If a KD mechanism of action section is added (suggested topics: glutamate/GABA balance, oxidative stress, inflammation, microbiome), and the major and minor comments below are addressed, this review could make a strong contribution to the literature.

Other major comments:

I think the paper would benefit greatly from a discussion of different types of ketogenic diet compositions. For example, carnivore and the Mediterranean ketogenic diet are two popular forms of genuine ketogenic diets rising in popularity among lay people and for clinical use. Obviously, while they could have the same macronutrient breakdown, their micronutrient compositions and fat sources vary. Such nuance should be reflected in the text, even if there is limited literature to cite. This comment is partially in response to the claim (line 105) that classic KD is mostly “saturated” LCTs.

The way the text is written implies that all KD and pseudo-KD diets are equal in efficacy. I suggest that prose is shifted to emphasize instead that there is a paucity of evidence. “Absence of evidence is not evidence of absence” etc. Perhaps this was the authors’ intention, but I think the messaging is important.

Furthermore, the authors themselves cite some evidence arguing for the superiority of 4:1 KD in some cases (e.g. 1. ref 30, LG syndrome 2. in lines 194-195 and 3. line 250) and imply, in the conclusion, that some patients might respond better to diets that induce higher BHB levels.

Minor comments:

In the abstract, the authors define KD/KDT as any diet the “would be expected to result in a ketogenic state.” I would contend a ketogenic diet is defined by the presence or absence of ketosis. If ketones are not >0.5 mM, it’s not a ketogenic diet. Period.

Please carefully review the text for formatting and terminology inconsistencies (just two e.g.s choose Krebs or TCA) or beta symbol in line 53 or “beta” text in line 61), typos, spacing issues, grammatical errors, etc. There are quite a few.

Line 59. Would not refer to ketosis as a “nutrient-deprived state.” This carries an overly negative connotation and is factually incorrect as glucose is the “nutrient” in question and glucose is non-essential in a dietary sense.

Line 109. Fasting is not an antiquated practice. It’s clinically very popular and even more so among the lay public for weight loss.

MCTs. Define more specifically, please. C8, pure caprylic? Virgin coconut oil?

MCTs. They can cross the BBB and be directly metabolized. This should be discussed as a possible mechanism of action and as a confounder when comparing BHB levels and correlation with efficacy in MCT diet vs. KD.

MAD. Line 211. “or fluids.” To what does this refer? Ketogenic diets do not restrict fluids in general.

MAD. Line 218. “all carbohydrates” should be revised to “carbohydrates irrespective of the glycemic index.” As written, it is confusing as the LGIT section is next.

Gastrointestinal. Please read Ho et al. 2012 “Stopping or Reducing Dietary Fiber Intake Reduces Constipation and Its Associated Symptoms” https://pubmed.ncbi.nlm.nih.gov/22969234/. Also, it’s worth noting that potassium citrate is severely constipating (line 372).

Cardiovascular. The authors must elaborate in this section as literature on KD and CV risk suffers from a lack of nuance in general. It may be helpful to read Norwitz and Loh 2020 “A Standard Lipid Panel Is Insufficient for the Care of a Patient on a High-Fat, Low-Carbohydrate Ketogenic Diet” https://pubmed.ncbi.nlm.nih.gov/32351962/

If the authors are going to comment on the rate of attrition on classic KD, they should discuss the general inadequacy of the system to support and educate patients and their families. Clinically, attrition is mostly due to a lack of information, not bona fide lack of palatability of the diet. Other studies (including the Virta two-year study) have demonstrated long-term adherence to a well-formulated diet is possible.

Author Response

Point 1:   The authors provide a good overview of the literature comparing different ketogenic or pseudo-ketogenic interventions in pediatric epilepsy. My most pressing concern is the manuscript does not, in its current form, add to the state of knowledge. KD for epilepsy is a century-old and vetted intervention and many prior reviews have made the similar points to those raised in this manuscript. Therefore, I suggest a supplementation of the existing document with a more comprehensive discussion of the possible mechanisms (emphasis on plural) that contribute to the anti-seizure effects of KDs. Such a hypothetical section would fit nicely before the conclusion and lead into the authors’ final well-made point that subcategorizing epilepsy cases based on metabolic signatures to provide personalized medicine is the future. If a KD mechanisms of action section is added (suggested topics: glutamate/GABA balance, oxidative stress, inflammation, microbiome), and the major and minor comments below are addressed, this review could make a strong contribution to the literature.

Response 1: Thank you for the suggestion.  The text was update to include potential mechanisms of action of ketogenic diet therapy. I agree this greatly enhances the paper and will make a strong contribution to the current body of literature.

Point 2:   I think the paper would benefit greatly from a discussion of different types of ketogenic diet compositions. For example, carnivore and Mediterranean ketogenic diet are two popular forms of genuine ketogenic diets rising in popularity among lay people and for clinical use. Obviously, while they could have the same macronutrient breakdown, their micronutrient compositions and fat sources vary. Such nuance should be reflected in the text, even if there is limited literature to cite. This comment is partially in response to the claim (line 105) that classic KD is mostly “saturated” LCTs.

Response 2: Thank you for this astute comment. The inclusion of the carnivore and Mediterranean ketogenic diets allowed us to expand upon the idea that KD fatty acid compositions vary greatly.  The text has been updated to reflect this idea.

Point 3: The way the text is written implies that all KD and pseudo-KD diets are equal in efficacy. I suggest that prose is shifted to emphasize instead that there is a paucity of evidence. “Absence of evidence is not evidence of absence” etc. Perhaps this was the authors’ intention, but I think the messaging is important.

Response 3: This is an important message.  The text has been updated to reflect the paucity of evidence in this subject matter.

Point 4:   Furthermore, the authors themselves cite some evidence arguing for the superiority of 4:1 KD in some cases (e.g. 1. ref 30, LG syndrome 2. in lines 194-195 and 3. line 250) and imply, in the conclusion, that some patients might respond better to diets that induce higher BHB levels.

Response 4: Thank you for your comment.  This idea has been incorporated into the “conclusion” portion of the text.

Point 5:   In the abstract the authors define KD/KDT as any diet the “would be expected to result in a ketogenic state.” I would contend a ketogenic diet is defined by the presence or absence of ketosis. If ketones are not >0.5 mM, it’s not a ketogenic diet. Period.

Response 5: Thank you for your insight, the text has been corrected

Point 6:   Please carefully review the text for formatting and terminology inconsistencies (just two e.g.s choose Krebs or TCA) or beta symbol in line 53 or “beta” text in line 61), typos, spacing issues, grammatical errors, etc. There are quite a few.

Response 6: Thank you for the observation.  We have corrected multiple errors in the article.  Please be advised spacing and citations may appear distorted due to track changes, formatting and use of citation software.  These will be corrected prior to publication.

Point 7:   Line 59. Would not refer to ketosis as a “nutrient-deprived state.” This carries an overly negative connotation and is factually incorrect as glucose is the “nutrient” in question and glucose is non-essential in a dietary sense.

Response 7: This does carry a negative connotation. The text was updated to include a “carbohydrate-limited state”

Point 8: Line 109. Fasting is not an antiquated practice. It’s clinically very popular and even more so among the lay public for weight loss.

Response 8: Thank you for your comment. The text was updated for clarification to: “Present day, initiating the KD with a fasting is considered an antiquated practice and many centers do not require a fast at all whatsoever.”

Point 9:   MCTs. Define more specifically please. C8, pure caprylic? Virgin coconut oil?

Response 9:  Thank you, the text was updated to read “Commercial MCT oils, generally isolated from coconut or palm oil...” for clarification. Medium chain fatty acid intermediates were also discussed further in the “mechanism” section.

Point 10:   MCTs. They can cross the BBB and be directly metabolized. This should be discussed as a possible mechanism of action and as a confounder when comparing BHB levels and correlation with efficacy in MCT diet vs. KD.

Response 10: Thank you for your comment, the mechanism of action for MCTs as well as the potential confounding with medium chain fatty acid intermediates has been addressed. 

Point 11:   MAD. Line 211. “or fluids.” To what does this refer to? Ketogenic diets do not restrict fluids in general.

Response 11: Thank you for the observation, “or fluids” has been removed from the text

Point 12:   MAD. Line 218. “all carbohydrates” should be revised to “carbohydrates irrespective of the glycemic index.” As written, it is confusing as the LGIT section is next.

Response 12: Thank you for your insight, the text has been updated.

Point 13:   Gastrointestinal. Please read Ho et al. 2012 “Stopping or Reducing Dietary Fiber Intake Reduces Constipation and Its Associated Symptoms” https://pubmed.ncbi.nlm.nih.gov/22969234/. Also, it’s worth noting that potassium citrate is severely constipating (line 372).

Response 13: Thank you for your comment and the article recommendation. Both points were included in the text.

Point 14:   Cardiovascular. The authors must elaborate in this section as literature on KD and CV risk suffers from a lack of nuance in general. It may be helpful to read Norwitz and Loh 2020 “A Standard Lipid Panel Is Insufficient for the Care of a Patient on a High-Fat, Low-Carbohydrate Ketogenic Diet” https://pubmed.ncbi.nlm.nih.gov/32351962/.

Response 14: Thank you for bringing this to our attention.  This is a very interesting topic and our text greatly benefits from its inclusion.

Point 15:   If the authors are going to comment on the rate of attrition on classic KD, they should discuss the general inadequacy of the system to support and educate patients and their families. Clinically, attrition is mostly due to a lack of information, not bona fide lack of palatability of the diet. Other studies (including the Virta two-year study) have demonstrated long-term adherence to a well-formulated diet is possible.

Response 15: Thank you for your comment and insight.  This idea has been updated in the text.

Round 2

Reviewer 2 Report

Your revisions are impressive and substantially improve the manuscript. I will recommend it for some minor edits. 

Line 23: to reflect changes elsewhere in the manuscript, please adapt to something like"there is limited evidence distinguishing among the efficacy of the four KDTs"

Line 71-74: important typo. as written, states fatty acids fuel the brain. I know this is now what you meant. Please change to BHB.

Line 76: BHB is oxidized to Acetoacetate in neurons and acetone is not a fuel, to the best of my knowledge. Please correct.

Line 94: suggestion to abbreviate classic ketotic diet and cKD to avoid confusion. KD can be read as including all four.

Line 228: Really? .8-1.6 mM? When I have patients on a 4:1, 3:1 they often are >2.0, some as high as 3.5.

Line 233: Just Mediterranean. It isn't specific to Spanish. Also, extra virgin olive oil, not just virgin. There's a difference. For interest, you may want to google the "extra virginity olive oil." Medit Keto also doesn't encourage red wine. carbs.

Line 252: Carnivore is not 0 carb. There are carbs in carnivore foods such as oysters, mussels, liver. Simply state all animal based.

Line 1092: suggestion to remove reference to glutamate as it almost contradicts the previous paragraph and there is research/clinical evidence on glutamate (free glutamate as MSG or in casein or gluten) exacerbating neurological diseases by throwing off GABA/glu balance.

Thank you for adding the suggested section. I think it greatly enhances the manuscript. 

Author Response

Point 1:   Line 23: to reflect changes elsewhere in the manuscript, please adapt to something like "there is limited evidence distinguishing among the efficacy of the four KDTs"

Response 1: Thank you for your comment, Line 23 and 24 were updated to “The composition of the 4 main KDTs differ, and limited evidence to distinguish the efficacy among different diets currently exists.”

Point 2:   Line 71-74: important typo. as written, states fatty acids fuel the brain. I know this is now what you meant. Please change to BHB.

Response 2: Thank you for identifying this error. After acceptance of initial track changes, this point changed to lines 60-61. The sentence was changed to “Ketone bodies, β-hydroxybutyrate (BHB) in particular, have been traditionally considered an alternative source of energy supply and metabolism in humans provides a significant source of fuel for the brain in a carbohydrate-limited state.

Point 3:   Line 76: BHB is oxidized to Acetoacetate in neurons and acetone is not a fuel, to the best of my knowledge. Please correct

Response 3: Thank you for this comment. For your reference, this point was changed to line 65-66. The sentence was edited to read: “During periods of very low carbohydrate intake, BHB is the primary energy source for neurons.”

Point 4:   Line 94: suggestion to abbreviate classic ketotic diet and cKD to avoid confusion. KD can be read as including all four.

Response 4: This is an excellent suggestion. Any reference to “classic” or “traditional” KD have been edited to cKD throughout the text.

Point 5:   Line 228: Really? .8-1.6 mM? When I have patients on a 4:1, 3:1 they often are >2.0, some as high as 3.5.

Response 5: Please see line 94 for this response.  The sentence was edited to read: “Achievement of a ketotic state is followed by a gradual introduction of the cKD over a 3-day period, working up to a KD ratio of 3:1 to 4:1 or until an adequate level of ketosis is achieved, generally a minimum of 80-160 mmol/L BHB.

Point 6:   Line 233: Just Mediterranean. Isn't specific to Spanish. Also, extra virgin olive oil, not just virgin. There's a difference. For interest you may want to google the "extra virginity olive oil." Medit Keto also doesn't encourage red wine. carbs.

Response 6: Thank you for your comment. Please see lines 99-104. “Spanish” was removed, and the diet is now referred to as the Mediterranean KD. The suggestion that red wine and carbohydrates are encouraged have been removed.

Point 7:   Line 252: Carnivore is not 0 carb. There are carbs in carnivore foods such as oysters, mussels, liver. Simply state all animal based.

Response 7: Thank you for the excellent point.  Please see that line 106 now reads “Another diet, the carnivore diet, encourages an animal-based meal pattern by eliminating all foods but meats, fish, and eggs.”

Point 8:   Line 1092: suggestion to remove reference to glutamate as it almost contradicts the previous paragraph and there is research/clinical evidence on glutamate (free glutamate as MSG or in casein or gluten) exacerbating neurological diseases by throwing off GABA/glu balance.

Response 8: Thank you for this suggestion.  The reference to glutamate was removed from line 456.

This manuscript is a resubmission of an earlier submission. The following is a list of the peer review reports and author responses from that submission.

Round 1

Reviewer 1 Report

The review is certainly accurate and extensive and reports complete data from recent literature. Unfortunately, the topic in recent years has been widely treated and the review does not bring significantly new data to what is already present in the literature. 

Author Response

Thank you for taking the time to review our recently submitted review article.  Please find our revisions and comments in the attached documents along with the inclusion of two tables to assist in clarification. I am very appreciative of the review process Nutrients completes prior to the publication to the journal.

I believe the comments have allowed us to improve our article for the readers and will assist in progressing research.

Reviewer 2 Report

Line 17: The preferred term is now antiseizure medication (ASM).

Line 25 This is a very interesting topic.  But one of the other confounding issues lies in individual responses to the diet-- is it type of epilepsy, a genetic marker, the microbiome, etc? It's probably more than the diet per se.

Line 46 It also related to the ease of use of the medications, rather than the restrictions of the diet.

Line 113 Historically the fast continued until the child lost 10% of body weight.  This evolved to 5% of BW, then multiple days, then finally a day at most.  Many centers now wouldn't fast at all.

Line 382  Sentence should probably be.  It will be important to identify KDTs that have the greatest efficacy.

Line 420  Probably need more references, discussion here.  Important issue.  See suggested additional references.

Line 519 Although there are certainly reports of people staying on it for protracted periods, presumably due to some sort of efficacy--- less seizures, better quality of life.

Line 541 As noted above.... there's more to the picture than mechanism of diet - it may be a story of more personalized medicine. The question may be  which mechanism in which individual patient.

These are some additional references that you might want to peruse and perhaps include:

Ketogenic Diet: A New Light Shining on Old but Gold Biochemistry.

Longo R, Peri C, Cricrì D, Coppi L, Caruso D, Mitro N, De Fabiani E, Crestani M.

Nutrients. 2019 Oct 17;11(10). pii: E2497. doi: 10.3390/nu11102497. Review

Ketogenic diet for schizophrenia: clinical implication.

Sarnyai Z, Kraeuter AK, Palmer CM.

Curr Opin Psychiatry. 2019 Sep;32(5):394-401. doi: 10.1097/YCO.0000000000000535.

PMID:

Dietary therapies: a worldwide phenomenon.

Kossoff EH, Caraballo RH, du Toit T, Kim HD, MacKay MT, Nathan JK, Philip SG.

Epilepsy Res. 2012 Jul;100(3):205-9. doi: 10.1016/j.eplepsyres.2011.05.024.

3119281

The Gut Microbiota Mediates the Anti-Seizure Effects of the Ketogenic Diet.

Olson CA, Vuong HE, Yano JM, Liang QY, Nusbaum DJ, Hsiao EY.

Cell. 2018 Jun 14;173(7):1728-1741.e13. doi: 10.1016/j.cell.2018.04.027. Epub 2018 May 24. Erratum in: Cell. 2018 Jul 12;174(2):497.

How does the ketogenic diet work? Four potential mechanisms.

Danial NN, Hartman AL, Stafstrom CE, Thio LL.

J Child Neurol. 2013 Aug;28(8):1027-33. doi: 10.1177/0883073813487598. Epub 2013 May 13.

PMID:23670253

The neuropharmacology of the ketogenic diet.

Hartman AL, Gasior M, Vining EP, Rogawski MA.

Pediatr Neurol. 2007 May;36(5):281-92. Review.

Does ketogenic diet have any negative effect on cardiac systolic and diastolic functions in children with intractable epilepsy?: One-year follow-up results.

Ozdemir R, Kucuk M, Guzel O, Karadeniz C, Yilmaz U, Mese T.

Brain Dev. 2016 Oct;38(9):842-7. doi: 10.1016/j.braindev.2016.03.009. Epub 2016 Apr 5.

PMID:27066714

The impact of the modified Atkins diet on lipid profiles in adults with epilepsy.

Cervenka MC, Patton K, Eloyan A, Henry B, Kossoff EH.

Nutr Neurosci. 2016;19(3):131-7. doi: 10.1179/1476830514Y.0000000162. Epub 2014 Nov 10.

PMID:25383724

Effects of ketogenic diet on vascular function.

Kapetanakis M, Liuba P, Odermarsky M, Lundgren J, Hallböök T.

Eur J Paediatr Neurol. 2014 Jul;18(4):489-94. doi: 10.1016/j.ejpn.2014.03.006. Epub 2014 Mar 25.

Cardiac complications in pediatric patients on the ketogenic diet.

Best TH, Franz DN, Gilbert DL, Nelson DP, Epstein MR.

Neurology. 2000 Jun 27;54(12):2328-30

Prospective study of growth and bone mass in Swedish children treated with the modified Atkins diet.

Svedlund A, Hallböök T, Magnusson P, Dahlgren J, Swolin-Eide D.

Eur J Paediatr Neurol. 2019 Jul;23(4):629-638. doi: 10.1016/j.ejpn.2019.04.001. Epub 2019 Apr 8.

1.

Potassium citrate and metabolic acidosis in children with epilepsy on the ketogenic diet: a prospective controlled study.

Bjurulf B, Magnus P, Hallböök T, Strømme P.

Dev Med Child Neurol. 2020 Jan;62(1):57-61. doi: 10.1111/dmcn.14393. Epub 2019 Nov 19.

Author Response

Response to Reviewer's Comments:

Point 1: Line 17: The preferred term is now antiseizure medication (ASM).

Response 1: The text has been corrected.

Point 2: Line 25 This is a very interesting topic.  But one of the other confounding issues lies in individual responses to the diet-- is it type of epilepsy, a genetic marker, the microbiome, etc? It's probably more than the diet per se.

Response 2: Thank you for your comment and insight. I agree, individual response is a confounding issue. We hope the inclusion of omics research may be able to assist us with identifying an appropriate diet for each individual. The text has been updated to reflect this topic.

Point 3: Line 46 It also related to the ease of use of the medications, rather than the restrictions of the diet.

Response 3: Thank you for your insightful comment, the text was updated to reflect this idea.

Point 4: Line 113 Historically the fast continued until the child lost 10% of body weight.  This evolved to 5% of BW, then multiple days, then finally a day at most.  Many centers now wouldn't fast at all.

Response 4: Thank you for your comment, the text was updated to reflect this point.

Point 5: Line 382  Sentence should probably be.  It will be important to identify KDTs that have the greatest efficacy.

Response 5: The text has been corrected.

Point 6: Line 420  Probably need more references, discussion here.  Important issue.  See suggested additional references.

Response 6: Thank you very much for your inclusion of additional references. These allowed me to improve upon the section and provide a better picture of the potential adverse effects of KDT and their impact on health outcomes.

Point 7: Line 519 Although there are certainly reports of people staying on it for protracted periods, presumably due to some sort of efficacy - less seizures, better quality of life.

Response 7: Thank you for your comment, the text was updated to reflect this point

Point 8: Line 541 As noted above.... there's more to the picture than mechanism of diet - it may be a story of more personalized medicine. The question may be which mechanism in which individual patient.

Response 8: Thank you for the astute observation. The idea of using metabolic profiling to identify metabolic signatures of seizures to form personalized nutrition interventions has been updated in the text. You correct, the question is not “what” mechanism, but “which” for each particular patient.

Reviewer 3 Report

Major Revisions

  • While the authors have reviewed human clinical trials, this manuscript lacks a discussion section.  There is no synthesis of new ideas, no hypotheses for future works or mechanisms of efficacy, and provides no additional information beyond the reporting of descriptive statistics from other manuscripts. What is this review adding to the current literature? It seems like just an overview of what’s already out there, which has already been done numerous times in numerous journals.
  •  “Relevance to this subject matter” is not a detailed description of methods. What criteria were used for “relevance”? Was subject age important? Sample size? Epilepsy type or severity?
  • What does “integrity of the information provided” mean?
  • There is largely too much focus on the % of people with each outcome within each study, which makes its hard to actually digest information. Summaries of the comparisons would be helpful rather than regurgitating descriptive statistics. A lot of this can be removed from the text and put into a table, and then the text could focus on more important things like the actual conclusions of studies and not overwhelming the reader with the results of each study.
  • The sentence “This would shift neurotransmitter pools and modulate membrane excitability, restoring the physiological balance of excitation and inhibition” on page 2 line 76 is unsupported. Please amend this or provide evidence of this. Though this is one broad hypothesis as to how neuronal activity is altered through keto-adaptation, this is currently an ongoing area of research with no underlying mechanisms known and should not be presented as a fact unless you have citations to support it.
  • The authors fail to mention that in many of the studies presented, children were not only on a ketogenic diet but were also on at least one other AED.
  • Although the authors report the types of side effects commonly reported, they do not discuss what this could mean or how it relates to potential outcomes. Furthermore, they do not discuss that the side effects could be related to patient conditions nor provide a comparison of side effects from healthy participants that would allow for the separation of the two influencing factors.
  • There is no discussion regarding the well-established fact that there are responders and non-responders to the ketogenic diet. Could the authors please comment on this and perhaps suggest how this influenced results of each study and potential reasoning?

Minor Revisions

  • The sentence “They are considered… nutrient-deprived state” on page 2 line 57 does not make sense.
  • Please define “RCTs” in abstract
  • The description of the classic ketogenic diet is outdated. Although the historical perspective is important, current applications of this diet do not typically include a 3 day fast, especially for children, so this should be noted. Also, weighing of the food is recommended at the start of each of these diets but the manuscript biases readers towards believing this restriction is limited to the KD alone.
  • The authors switch between using numbers and writing out the words (ie sometimes use “6” and sometimes use “six”). Please pick one and maintain throughout to enhance readability. (Example: “At three months, 11 children … with five children” page 4 line 143).
  • What is the Z score referring to on the top of page 5? Z score of the infant weight? Seizures?
  • The portion of the discussion section with the figures on retention would be better served as a subsection of the results.
  • Within the same paragraph the authors ask whether mechanism-identification is beneficial and also assert that it is “imperative” and “necessary”. Why ask this question if you’re going to state the answer as an asserted fact?

Author Response

Response to Reviewer's  Comments:

Point 1: While the authors have reviewed human clinical trials, this manuscript lacks a discussion section.  There is no synthesis of new ideas, no hypotheses for future works or mechanisms of efficacy, and provides no additional information beyond the reporting of descriptive statistics from other manuscripts. What is this review adding to the current literature? It seems like just an overview of what’s already out there, which has already been done numerous times in numerous journals.

Response 1: Thank you for your insightful comment.  We appreciate your feedback.  This review article has been updated significantly to discuss the possibility of isolating metabolic signatures of seizures in order to provide individualized nutrition interventions. We believe this review article will encourage researchers to study the novel idea of a targeted approach for patients rather than blanket restrictive diet therapies. In the future, targeted approaches with individualized nutrition interventions may improve the quality of lives for children and adolescents with refractory epilepsy by decreasing seizure frequency and side effects and promoting growth.

Point 2: “Relevance to this subject matter” is not a detailed description of methods. What criteria were used for “relevance”? Was subject age important? Sample size? Epilepsy type or severity?

Response 2: Thank you for your observation and comment, the text was updated to reflect this point.

Point 3: What does “integrity of the information provided” mean?

Response 3: Thank you for your comment, this phrase was removed to ensure clarity.

Point 4: There is largely too much focus on the % of people with each outcome within each study, which makes its hard to actually digest information. Summaries of the comparisons would be helpful rather than regurgitating descriptive statistics. A lot of this can be removed from the text and put into a table, and then the text could focus on more important things like the actual conclusions of studies and not overwhelming the reader with the results of each study.

Response 4: Thank you for your insight and comment. I have taken this under serious consideration and have attempted to summarize the research and add tables in order to make the text easier to digest.

Point 5: The sentence “This would shift neurotransmitter pools and modulate membrane excitability, restoring the physiological balance of excitation and inhibition” on page 2 line 76 is unsupported. Please amend this or provide evidence of this. Though this is one broad hypothesis as to how neuronal activity is altered through keto-adaptation, this is currently an ongoing area of research with no underlying mechanisms known and should not be presented as a fact unless you have citations to support it.

Response 5: Thank you for your comment, this phrase was removed in order to not present the mechanism as fact.

Point 6: The authors fail to mention that in many of the studies presented, children were not only on a ketogenic diet but were also on at least one other AED.

Response 6: This is an important point, thank you for bringing it to our attention. The text was updated to reflect this.

Point 7: Although the authors report the types of side effects commonly reported, they do not discuss what this could mean or how it relates to potential outcomes. Furthermore, they do not discuss that the side effects could be related to patient conditions nor provide a comparison of side effects from healthy participants that would allow for the separation of the two influencing factors.

Response 7: Thank you for your comment.  The text has been updated to reflect our concern for the impact of adverse effects on short- and long-term growth. We also discuss how individuals with adverse effects are likely to discontinue the diet, making it difficult for researchers to evaluate the true impact of KDT on health outcomes over time.

Point 8: There is no discussion regarding the well-established fact that there are responders and non-responders to the ketogenic diet. Could the authors please comment on this and perhaps suggest how this influenced results of each study and potential reasoning?

Response 8: This is an important concept, thank you for bringing it to our attention. The concept of responders and non-responders has been added in the section 3.6.6 and in the discussion.

Point 9: The sentence “They are considered… nutrient-deprived state” on page 2 line 57 does not make sense.

Response 9: Thank you for the observation, the sentence was corrected in the text.

Point 10: Please define “RCTs” in abstract

Response 10: The text has been corrected.

Point 11: The description of the classic ketogenic diet is outdated. Although the historical perspective is important, current applications of this diet do not typically include a 3 day fast, especially for children, so this should be noted. Also, weighing of the food is recommended at the start of each of these diets but the manuscript biases readers towards believing this restriction is limited to the KD alone.

Response 11: The text has been updated to report current classic ketogenic diets do not typically start with a 3-day fast.  The text has also been updated to report weighing food is common at the beginning of KDTs.

Point 12: The authors switch between using numbers and writing out the words (ie sometimes use “6” and sometimes use “six”). Please pick one and maintain throughout to enhance readability. (Example: “At three months, 11 children … with five children” page 4 line 143).

Response 12: Thank you for your comment, the text was updated to maintain consistency.

Point 13: What is the Z score referring to on the top of page 5? Z score of the infant weight? Seizures?

Response 13: Yes, this would be the Z score of the infant weight. This was reflected in the text.

Point 14: The portion of the discussion section with the figures on retention would be better served as a subsection of the results.

Response 14: This is an excellent suggestion, the text has been updated accordingly.

Point 15: Within the same paragraph the authors ask whether mechanism-identification is beneficial and also assert that it is “imperative” and “necessary”. Why ask this question if you’re going to state the answer as an asserted fact?

Response 15: Thank you for bringing this to our attention. The assertion has been removed and replaced with thoughtful consideration.

Round 2

Reviewer 1 Report

None

Reviewer 3 Report

The authors failed to address many of the major concerns for the manuscript, even when they have indicated that they have done so. Most notably, the discussion section still does not include an actual discussion of the significance of the data provided.

Points 2 & 3 were not addressed in the slightest. 

Point 4 was not taken seriously. A woefully under-representative table was included (within the discussion section?) while descriptive statistics continue to be the mainstay of the results section without meaningful summarization. 

etc.